# Influencing Factors on the Household-Waste-Classification Behavior of Urban Residents: A Case Study in Shanghai

**DOI:** 10.3390/ijerph19116528

**Published:** 2022-05-27

**Authors:** Decai Tang, Lei Shi, Xiaojuan Huang, Ziqian Zhao, Biao Zhou, Brandon J. Bethel

**Affiliations:** 1School of Law and Public Affairs, Nanjing University of Information Science & Technology, Nanjing 210044, China; tangdecai@nuist.edu.cn (D.T.); 20191227003@nuist.edu.cn (L.S.); 2School of Law and Business, Sanjiang University, Nanjing 210012, China; 3China Institute of Manufacturing Development, Nanjing University of Information Science & Technology, Nanjing 210044, China; 4School of Business, Jiangsu Open University, Nanjing 210000, China; 5School of Foreign Languages, Nangjing University of Finance and Economics, Nanjing 210023, China; 9119901022@nufe.edu.cn; 6School of Marine Sciences, Nanjing University of Information Science & Technology, Nanjing 210044, China; 20195109101@nuist.edu.cn

**Keywords:** household waste classification, influencing factors, empirical analysis

## Abstract

As the process of urbanization in China continues to accelerate, the amount of domestic waste generated correspondingly increases and directly affects the living space of residents. This indirectly implies that to reduce the production of municipal solid waste and the need for garbage disposal and recycling, household-waste-classification activities by the residents are of great significance. Using Shanghai as a case study, this study investigated the influencing factors on residents’ household waste classification by conducting a survey. Statistical analysis was then adopted, which is specified below. First, this study proposed research hypotheses related to the influencing factors of residents’ domestic-waste-sorting behavior from three levels: government, society and individuals. Second, the study designed a questionnaire from five perspectives: individual characteristic variables, government, society, residents and classification behavior. Then, SPSS software was used to carry out descriptive statistical, reliability and validity assessments using ANOVA, correlation and regression analyses on the sample data obtained from the questionnaire. The results suggested that the research hypotheses were statistically significant: (1) females and residents with higher education were more likely to participate in domestic waste classification; (2) reward and punishment measures had the most significant impact on residents’ waste-classification behavior; and (3) publicity and education, classification standards, classification facilities, the recycling system, subjective norms, environmental knowledge and environmental attitudes all had a positive effect on residents’ household waste classification. Finally, based on the results of the empirical analysis, this paper provides reference suggestions for the further development of domestic waste classification in Shanghai.

## 1. Introduction

With the continuous acceleration of China’s modernization process and the rapid improvement of people’s living standards, the quantity and types of domestic waste have gradually increased. According to the 2020 National Annual Report on the Prevention and Control of Environmental Pollution by Solid Waste in Large and Medium Cities released by the Ministry of Ecology and Environment of China, the total production of domestic waste in 196 large- and medium-sized cities in China totaled 235.602 million tons. Among them, Shanghai produced the largest amount of domestic waste, accounting for 10.768 million tons, or approximately 4.57%. The extremely large quantity of waste produced has evolved into a common problem faced throughout the country, and as such, has strongly affected the cities’ hygiene and residents’ health. To meet the growing demand for a high-quality living environment, domestic waste classification was proposed to reduce this waste from its source, thereby improving treatment efficiency at later stages and realizing resource utilization of municipal solid waste.

From the residents’ perspectives, the classification and management of domestic waste were deeply analyzed, and the influencing factors of residents’ implementation of household waste classification were thoroughly understood. Subsequently, the problems existing in the process of promoting waste classification in Shanghai were drawn, the reasons were clarified and corresponding solutions were proposed, which will inevitably provide a decision-making reference for the orderly development of waste classification in Shanghai.

## 2. Research Hypothesis

The influencing factors of residents’ source classification and release of municipal solid waste are diverse, including not only the impact on the external environment but also the subjective factors of residents. When it comes to external factors, the most popular one is policy and incentives, which were shown to substantially influence residents’ waste-sorting behavior [1,2,3]. However, internal factors, such as psychological constructs, can also produce an essential influence on people’s waste-separation intention [4]. However, to the best of our knowledge, there is no piece of work that investigated both the outer and inner factors of people’s waste-sorting behavior in Shanghai, the city in China that pioneered in launching an official waste-sorting campaign. In this regard, this research contributed to filling in such a gap. According to previous research, the influencing factors of household-waste-classification behavior of Shanghai residents were hypothesized from three levels: government, society and residents.

### 2.1. Governmental Factors

A.Impact of publicity and education on residents’ household-waste-classification behavior

In the process of promoting household waste classification in various countries, publicity and education are widely used as basic means. Rousta et al. [5], Liu et al. [6], Choon et al. [7] and Sarbassov et al. [8] investigated the household-waste-classification behavior of Swedish, Chinese, Malaysian and Kazakh residents, respectively. The results suggest that when government departments publicize the relevant contents of waste classification to residents, residents’ awareness of participating in waste classification can be effectively enhanced and the implementation of waste-classification behavior by residents can be effectively promoted. Cui et al. [9] researched waste sorting in Beijing, China. They determined in the study that in order to carry out household waste classification work well, the relevant government departments should strengthen publicity, enrich publicity means, improve publicity facilities, and innovate perspectives and methods in the publicity process to improve residents’ acceptance of classification knowledge, thereby improving residents’ enthusiasm to participate in waste classification. Therefore, the following hypothesis was put forward:

**Hypothesis** **1a** **(H1a).**
*Publicity and education have a positive effect on residents*
*’ household-waste-classification behavior. In other words, the greater the publicity intensity and the more various the forms of publicity, the more likely residents are to participate in household-waste-classification activities.*


B.Impact of classification criteria on residents’ household-waste-classification behavior

Zheng et al. [10] and Wang [11] verified that the household-waste-classification standard is related to whether residents can understand and easily implement waste classification. Scientific and reasonable classification standards can promote residents’ waste-classification awareness and improve their enthusiasm to participate in waste classification. Therefore, the following hypothesis was put forward:

**Hypothesis** **1b** **(H1b).**
*The classification standard has a positive effect on residents’ household-waste-classification behavior. In other words, the more reasonable and understandable the classification criteria are, the more likely residents are to participate in household-waste-classification activities.*


C.Impact of reward and punishment measures on residents’ household waste classification

Lucia et al. [12], Guo et al. [13], Miafodzyeva et al. [14], Convery et al. [15] and Wu [16] claimed that residents’ household waste classification will be significantly affected by reward and punishment policies, and positive economic incentives are more easily accepted by residents. Wadehra et al. [17] confirmed in the investigation of waste classification in India that the quality of the reward and punishment mechanism is closely related to the implementation effect of the classification policy, and a high-quality reward and punishment mechanism can strengthen the residents’ willingness to classify garbage and promote the residents to implement the behavior of waste classification. Therefore, the following hypothesis was put forward:

**Hypothesis** **1c** **(H1c).**
*Reward and punishment measures have a significant positive effect on residents’ household-waste-classification behavior. In other words, the greater the rewards and punishments, the more likely residents are to participate in household-waste-classification activities.*


### 2.2. Social Factors

A.Impact of classification facilities on residents’ household waste classification

According to surveys of residents’ willingness to classify waste, Liu et al. [18], Malmir et al. [19], Wan et al. [20] and Kirakozian [21] showed that waste-classification infrastructure will have an impact on residents’ willingness to participate in household waste classification, and its high quality, including convenience, is positively correlated with residents’ enthusiasm for participating in waste classification. Zhang et al. [22] reported that low-quality waste-collection facilities will significantly reduce residents’ willingness to classify garbage when studying the current situation of the garbage sorting and recycling system in Chengdu. Therefore, the following hypothesis was put forward:

**Hypothesis** **2a** **(H2a).**
*Classification-supporting facilities play a positive role in promoting residents’ household-waste-classification behavior. In other words, the higher the quality of the classification facilities, particularly, the higher the convenience, the more likely residents are to participate in household-waste-classification activities.*


B.Impact of the recycling system on the household waste classification of residents

Cui et al. [9] further standardized the process of waste collection and showed that this can effectively strengthen the willingness of residents to classify garbage. Vassanadumrongdee et al. [23] and Meng et al. [24] insisted on standardizing the recycling, transportation and disposal of the garbage that is sorted by residents. It was suggested that this is very important for enhancing the enthusiasm of residents to participate in waste classification. If residents find that the sorted garbage is not being processed, it will reduce their enthusiasm and enthusiasm for continued classification, thus reducing their willingness to participate in waste classification. Therefore, the following hypothesis is put forward:

**Hypothesis** **2b** **(H2b).**
*The recycling system has a positive promoting effect on residents’ household-waste-classification behavior. In other words, the more standardized the recycling system is, the more likely residents are to participate in household-waste-classification activities.*


### 2.3. Resident Factors

A.Impact of subjective norms on residents’ household waste classification

Subjective norms refer to other individuals or organizations that are important to residents, such as family, friends, neighbors, colleagues, the government and environmental protection associations, whose attitudes and behaviors have a profound impact on individual residents. It includes the subject’s specific perception of the passive pressure of public opinion and the subjective will to cater to the expectations of the public opinion. Shaufique et al. [25] showed in their research on garbage recycling in Minnesota that social pressure will have an impact on individual-specific behavior decisions, that is, the expectations and views of other important individuals or organizations around individuals often affect the willingness of residents to participate in household waste classification. Janmaimool [26] and Wang et al. [27] came to a similar conclusion that subjective norms significantly affect residents’ garbage-sorting behavior. Therefore, the following hypothesis was put forward:

**Hypothesis** **3a** **(H3a).**
*Subjective norms play a positive role in promoting residents’ household-waste-classification behavior. In other words, the stronger the residents’ subjective perception of the expectations of the social reference groups and the higher the degree of compliance, the more likely they are to participate in household waste classification.*


B.Impact of environmental knowledge on residents’ household waste classification

Márquez et al. [28], Babaei et al. [29] and Almasi et al. [30] carried out investigations and studies on the waste classification influencing factors of Mexican and Iranian residents. Those studies pointed out that improving residents’ knowledge of waste classification can effectively enhance residents’ willingness to participate in waste classification. Therefore, the following hypothesis was put forward:

**Hypothesis** **3b** **(H3b).**
*Environmental knowledge has a positive effect on residents’ household-waste-classification behavior. In other words, the richer the residents’ environmental knowledge, the more likely they are to participate in household-waste-classification activities.*


C.Impact of environmental attitudes on residents’ household waste classification

Mahmud et al. [31] and Pakpour et al. [32] found in their study of waste-classification influencing factors of Malaysian and Iranian residents that residents’ attitudes and views on waste classification have an indirect impact on their classification intention. This is supported by Rauwald et al. [33] and Li et al. [34], who found that residents’ views on waste classification have a direct impact on their willingness to classify waste. Therefore, the following hypothesis was put forward:

**Hypothesis** **3c** **(H3c).**
*A pro-environmental attitude has a positive effect on residents’ household-waste-classification behavior. In other words, the more positive the residents’ attitude toward environmental protection, the more likely they are to participate in household-waste-classification activities.*


## 3. Questionnaire Design

It was validated that a survey is one of the best ways to obtain first-hand data for studies concerning waste management [35,36].

Based on the above research hypotheses, a total of 42 questions were designed to investigate household-waste-classification activities in this study. The concise questionnaire formulation process is clarified below in this section. All questions originated from previous studies and were modified to fit into this research context. In total, 700 questionnaires were distributed through online and offline channels in the urban area of Shanghai city, which ended up producing 517 valid electronic and 106 valid paper questionnaires. Online questionnaires were distributed through www.wjx.cn (accessed on 12 October 2021), which is the most popular online questionnaire design, distribution and collection website and could reach the most representative respondents of this study. Offline paper questionnaires were mainly distributed to the elderly who had difficulties accessing the internet. Several neighborhoods that had high numbers of elderly in Shanghai urban districts were selected. To avoid sample bias, the number of questionnaires that were distributed to different districts was varied according to the population size. The research team was comprised of four people from the university’s research lab that conducted this survey. Two months and one week were used for questionnaire pre-test, modification and official distribution.

In the first section of this questionnaire, questions were set to investigate the respondents’ socio-demographic characteristics, such as gender (i.e., male or female), age, level of education (i.e., senior high school or below, junior college, bachelor’s, master’s or above) and profession (i.e., student, self-employed, education workers, government staff, retired worker, company employee or other).

The independent variables were set according to the three levels of government, society and residents, and 31 terms were designed. According to the Likert scale, five responses were designed: “strongly agree”, “agree”, “neutral”, “disagree” and “strongly disagree”. The dependent variable was the construct of household-waste-classification behavior, which was comprised of seven items (questions). There were five options for each question, which were “always classified”, “often classified”, “occasionally classified”, “improbable classified” and “never classified”.

### 3.1. Government

From the point of view of the government, three independent variables were set, namely, publicity and education, classification standards, and measures of reward and punishment (Table 1, Table 2 and Table 3).

**H1a.** 
*Publicity and education have a positive effect on residents’ household-waste-classification behavior; in other words, the greater the publicity intensity and the more abundant the publicity forms, the more likely residents are to participate in household-waste-classification activities. H1a was examined using four item dimensions.*


**H1b.** 
*The classification standard has a positive effect on residents’ household-waste-classification behavior. In other words, the more reasonable and understandable the classification criteria are, the more likely residents are to participate in household-waste-classification activities. H1b was examined using three item dimensions.*


**H1c.** 
*Reward and punishment measures have a significant positive effect on residents’ household-waste-classification behavior. In other words, the greater the rewards and punishments, the more likely residents are to participate in household-waste-classification activities. H1c was examined using four item dimensions.*


### 3.2. Social

Two independent variables were set up from a social point of view, namely, classified supporting facilities and recycling systems (Table 4 and Table 5).

**H2a.** 
*Classified supporting facilities play a positive role in promoting residents’ household-waste-classification behavior. In other words, the higher the quality of the classification facilities and the higher the convenience, the more likely residents are to participate in household-waste-classification activities. H2a was examined using four item dimensions.*


**H2b.** 
*The recycling system has a positive promoting effect on residents’ household-waste-classification behavior. In other words, the more standardized the recycling system is, the more likely residents are to participate in household-waste-classification activities. H2b was examined using four item dimensions.*


### 3.3. Residents

Three independent variables were set from the perspective of residents, namely, subjective norms, environmental knowledge and environmental attitudes (Table 6, Table 7 and Table 8).

**H3a.** 
*Subjective norms play a positive role in promoting residents’ household-waste-classification behavior. In other words, the stronger the residents’ subjective perception of the expectations of the social reference groups and the higher the degree of compliance, the more likely they are to participate in household waste classification. H3a was examined using four item dimensions.*


**H3b.** 
*Environmental knowledge has a positive effect on residents’ household-waste-classification behavior. In other words, the richer the residents’ environmental knowledge, the more likely they are to participate in household-waste-classification activities. H3b was examined using four item dimensions.*


**H3c.** 
*Environmental attitude has a positive effect on residents’ household-waste-classification behavior. In other words, the more positive the residents’ attitude toward environmental protection, the more likely they are to participate in household-waste-classification activities. H3c was examined using four item dimensions.*


### 3.4. Classification Behavior

In this study, the household-waste-classification behavior of residents was used as a dependent variable to evaluate the classification of household waste. Taking the implementation level of household waste classification as the model’s dependent variable, the questionnaire presented five options: “always classified”, “often classified”, “occasionally classified”, “improbable classified” and “never classified” (Table 9).

## 4. Data Analysis

A combination of electronic and on-site questionnaires was used in this study. A total of 517 valid electronic and 106 valid paper questionnaires were collected, totaling 623 valid questionnaires. This section comprises the results of the descriptive analysis, reliability and validity analysis of the scale, difference analysis, correlation analysis and regression analysis.

### 4.1. Descriptive Statistical Analysis

First, this study analyzed the basic characteristics of the valid questionnaires and analyzed the respondents according to their personal trait variables (gender, age, education level, occupation, etc.), reflecting the suitability of the questionnaire coverage.

As shown in Table 10, in terms of gender, there were slightly more males (51.7%) than females (48.3%). In terms of the age structure, the largest number of respondents were aged between 35 and 55, accounting for 50.4% of the total number of respondents, followed by those aged between 19 and 35, accounting for 28.6% of the total number of respondents. In terms of education level, the highest proportion of respondents had a bachelor’s degree (45.4%), followed by a junior college degree (27.9%) and a master’s degree or above (14.0%). In terms of occupation, the highest percentage of respondents were company employees (40.0%), followed by education workers (17.8%), and the remaining occupations were relatively evenly distributed. In general, the demographic characteristics of the valid sample in this study are relatively evenly distributed and representative.

### 4.2. Reliability Analysis

A reliability analysis was performed on the questionnaire to guarantee the consistency and stability of the data. In this study, the reliability of the questionnaire was tested by using the reliability coefficient method of Cronbach’s α. The reliability of a questionnaire is generally considered to be very high when Cronbach’s α is greater than 0.9. When Cronbach’s α is greater than 0.7 but less than 0.9, the reliability of the questionnaire is high. When Cronbach’s α is greater than 0.6 but less than 0.7, the reliability of the questionnaire is acceptable. If Cronbach’s α is less than 0.6, it means that the reliability of the questionnaire is poor and that the questionnaire needs to be revised and more data needs to be collected in a new survey [46]. In this study, the reliability analysis was conducted on the data of each variable of the sample separately. As shown in Table 11, the Cronbach’s α values for the governmental, social and resident factors, as well as the waste-sorting behavior, were all greater than 0.8. This indicated that the reliability of the questionnaire in this study was relatively high and the questionnaire could be analyzed empirically.

### 4.3. Validity Analysis

The validity of the variables (publicity and education, sorting standards, reward and punishment measures, auxiliary facilities for sorting, recycling system, subjective regulations, environmental knowledge, environmental attitudes, sorting behavior) was tested by using structural validity analysis. As shown in Table 12, the KMO value of the sample was found to be 0.873, which is greater than 0.6; the chi-squared value of Bartlett’s spherical test was 4932.868; the degree of freedom was 149; and the significance was 0.000. This indicated that the variables were correlated, the variables were set reasonably and the questionnaire was valid.

### 4.4. Difference Analysis

In this study, the differences in waste-sorting behavior between respondents with different demographic characteristics were analyzed, and the test procedure and results are shown below.

#### 4.4.1. Analysis of Differences in Waste-Sorting Behavior between Respondents of Different Genders

Differences in the waste-sorting behavior of respondents of different genders were analyzed using the independent sample t-test. As shown in Table 13, there was a significant difference in waste-sorting behavior between respondents of different genders (t = −2.574, *p* < 0.05), indicating that females were more likely to sort household waste than males.

#### 4.4.2. Analysis of Differences in Waste-Sorting Behavior between Respondents of Different Ages

One-way ANOVA was used to test for differences in waste-sorting behavior among respondents of different ages. As shown in Table 14, there was no significant difference in waste-sorting behavior between respondents of different ages (F = 1.521, *p* > 0.05), indicating that there was no significant effect of age on respondents’ waste-sorting behavior.

#### 4.4.3. Analysis of Differences in Waste-Sorting Behavior between Respondents with Different Education Levels

One-way ANOVA was used to test for differences in waste-sorting behavior between respondents with different education levels. As shown in Table 15, there was a significant difference in waste-sorting behavior among the respondents with different education levels (F = 7.644, *p* < 0.05). The higher the education level, the more likely the respondents were to engage in waste sorting.

#### 4.4.4. Analysis of Differences in Waste-Sorting Behavior between Respondents of Different Occupations

One-way ANOVA was used to test for differences in waste-sorting behavior between respondents of different occupations. As shown in Table 16, there was no significant difference in waste-sorting behavior between respondents with different occupations (F = 1.952, *p* > 0.05), indicating that there was no significant effect of occupation on respondents’ waste-sorting behavior.

### 4.5. Correlation Analysis

First, the correlation between governmental, social and residential factors and the respondents’ household waste-sorting behavior was initially investigated using Pearson’s correlation analysis, and the results of the study are shown in Table 17. In terms of the governmental factors, publicity and education (r = 0.522, *p* < 0.01), sorting standards (r = 0.548, *p* < 0.01), and reward and punishment measures (r = 0.562, *p* < 0.01) were significantly and positively correlated with respondents’ waste-sorting behavior. Among the social factors, auxiliary facilities for sorting (r = 0.508, *p* < 0.01) and recycling systems (r = 0.525, *p* < 0.01) were significantly and positively correlated with respondents’ waste-sorting behavior. Among the resident factors, subjective regulation (r = 0.515, *p* < 0.01), environmental knowledge (r = 0.509, *p* < 0.01) and environmental attitude (r = 0.477, *p* < 0.01) were significantly and positively correlated with the respondents’ waste-sorting behavior.

### 4.6. Regression Analysis

In order to further investigate the influence of governmental, social and residential factors on the respondents’ waste-sorting behavior, a multivariate regression analysis was carried out with publicity and education, sorting standards, reward and punishment measures, auxiliary facilities for sorting, recycling system, subjective regulation, environmental knowledge and environmental attitude as independent variables, and waste-sorting behavior as the dependent variable.

A multivariate regression equation can be expressed as:$$Y=\beta_{0}+\beta_{1}X_{1}+\beta_{2}X_{2}+\ldots +\beta_{p}X_{p}+\varepsilon$$

*Y* is the dependent variable; *X*_1_, …, *X_p_* are the independent variables; $\beta_{0}$ is the intercept; $\beta_{1}$, …, $\beta_{p}$ are the estimated coefficients; and ε is the random error.

As shown in Table 18, from the model summary, the R^2^ of the model was 0.535, indicating that publicity and education, sorting standards, reward and punishment measures, auxiliary facilities for sorting, recycling system, subjective regulation, environmental knowledge and environmental attitude could predict 53.5% of the variance in waste-sorting behavior, and overall, the explanatory power of the model was fair. As shown in Table 19, from the ANOVA results of the model, the F-value was 88.356 (*p* < 0.001), indicating a significant linear relationship between the independent variables and the dependent variable of the model in this study.

The regression coefficients of the model are shown in Table 20. There, it can be found that the standardized regression coefficient of publicity and education on waste-sorting behavior was 0.143 (*p* < 0.01) in terms of governmental factors. This indicated that the implementation of publicity and education activities was conducive to motivating and promoting residents to engage in household-waste-sorting behavior. Moreover, the greater the publicity efforts and the richer the forms, the more likely the residents were to participate in household waste-sorting activities. Therefore, H1a of this study was true. The standardized regression coefficient of sorting standards on waste-sorting behavior was 0.155 (*p* < 0.01), indicating that sorting standards had a positive effect on encouraging and promoting residents’ household waste-sorting behavior. In other words, the more reasonable the sorting standards were and the more easily understood they were, the more likely the residents were to participate in household-waste-sorting activities. Therefore, H1b of this study was true. The standardized regression coefficient of reward and punishment measures on waste-sorting behavior was 0.181 (*p* < 0.01), indicating that the implementation of reward and punishment measures was conducive to promoting household-waste-sorting behavior among residents. That is, the stronger the reward and punishment measures, the more likely the residents were to participate in household-waste-sorting activities. Therefore, H1c of this study was true. The three influencing factors, in order of weight proportion, were reward and punishment measures (0.157), sorting standards (0.131), and publicity and education (0.121).

In terms of social factors, the standardized regression coefficient of auxiliary facilities for sorting on waste-sorting behavior was 0.100 (*p* < 0.01), indicating that auxiliary facilities for sorting had a positive role in stimulating and promoting household-waste-sorting behavior among residents. In other words, the more complete and convenient the auxiliary facilities for sorting were, the more likely the residents were to participate in household-waste-sorting activities. Therefore, H2a of this study was true. The standardized regression coefficient of a recycling system on waste-sorting behavior was 0.089 (*p* < 0.05), which indicated that a recycling system was conducive to encouraging and promoting household-waste-sorting behavior among residents. In other words, the more normative the recycling system was, the more likely the residents were to participate in household-waste-sorting activities. Thus, H2b of this study was verified. The influencing factors, in order of weight, were a recycling system (0.082) and auxiliary facilities for sorting (0.079).

In terms of residential factors, the standardized regression coefficient of subjective regulation on waste-sorting behavior was 0.134 (*p* < 0.01), indicating that subjective regulation was conducive to promoting and facilitating the household-waste-sorting behavior of residents. In other words, the higher the residents’ subjective perception of the expectations of the social reference group and the higher the degree of compliance, the greater the likelihood that residents engaged in household waste sorting. Therefore, H3a was considered validated. The standardized regression coefficient of environmental knowledge on waste-sorting behavior was 0.122 (*p* < 0.01), indicating that environmental knowledge had a positive role in encouraging and promoting the household-waste-sorting behavior of residents. In other words, the more environmental knowledge residents had, the more likely they were to participate in household-waste-sorting activities. Therefore, H3b was verified. The standardized regression coefficient of environmental attitude on waste-sorting behavior was 0.093 (*p* < 0.01), indicating that environmental attitude had a positive role in stimulating and promoting the household-waste-sorting behavior of the residents. In other words, the more positive the residents’ attitude toward environmental protection, the more likely they were to participate in household-waste-sorting activities. Therefore, H3c was true. The three influencing factors, in order of weight, were subjective regulation (0.122), environmental knowledge (0.116) and environmental attitude (0.074).

## 5. Discussion and Conclusions

Considering the importance of waste classification in medium and large cities, this study investigated questionnaire responses from residents of Shanghai. Collecting 637 valid samples in total, this study produced the following results and arguments after statistical analyses.

With regard to the socio-demographic characteristics and waste-classification behavior of residents, the results suggested that females and people with higher education tended to be more willing to sort waste, which are consistent with previous studies [1]. It was identified that women were more likely to participate in household waste sorting than men, which may be related to the fact that women undertake more housework. Additionally, the higher the education level of residents, the higher the likelihood of their participation in waste sorting. In the process of education, residents can receive relevant knowledge about waste sorting. The longer they are educated, and the higher their education level is, the more environmental knowledge they will receive and the more conducive it will be for residents to engage in household-waste-sorting activities.

When it came to the independent variables of the government, society and residents, the results are presented below.

The governmental factors that influenced residents’ waste-sorting behavior were, in order of weight, reward and punishment measures, sorting standards, and publicity and education. This suggested that economic means could significantly promote residents’ waste-sorting behavior, and that relevant government departments should introduce relevant policies, improve reward and punishment measures, and set reasonable and easily understood sorting standards, which were validated multiple times in previous research [48,49,50,51]. The effect of publicity and education on residents’ waste-sorting behavior works for a long period after its implementation, and a permanent mechanism should be established.

Socially, the factors that influenced residents’ waste-sorting behavior were, in order of weight, the recycling system and supporting facilities for sorting. The standardization of the recycling system enabled residents to feel that it was meaningful to sort their waste, which, in turn, could effectively increase their motivation to sort their waste from the source. Therefore, the sanitation department and the waste-recycling company should standardize the operations of the whole process, improve the supporting facilities for waste sorting and motivate residents to sort at the source.

For residents, the factors that influenced residents’ waste-sorting behavior were, in order of weight, subjective norms, environmental knowledge and environmental attitudes. We live in a society and are influenced by people around us all the time. The waste-sorting behavior of the people around us will create public opinion pressure on the residents themselves, forcing them to engage in waste-sorting activities. The community can mobilize the power of the masses to enable everyone to engage in household waste sorting, which is led by Chinese Communist Party members and officials, guided and monitored by volunteers, and participated in by residents. At the same time, the community should step up its efforts to publicize the environmental knowledge in the residential area, guide the residents to become actively involved in the public affairs of the community, cultivate the residents’ sense of ownership and allow them to participate in the waste-sorting work with a positive attitude.

This study innovated in terms of exploring the external and internal factors of waste-sorting behavior of Shanghai residents. Furthermore, the results demonstrated that government, society, and resident’s environmental attitude and knowledge could also influence their intention of waste classification. Moreover, publicity also played a very important role in promoting the public’s waste-sorting recognition, which should be rolled out broadly, covering primary students.

However, there were limitations to this study. First, this study did not dig deeper into how the three independent variables interact with each other and synergistically exert effects on residents’ sorting behavior. Second, the COVID-19 pandemic may have produced impacts on people’s willingness toward waste separation; more studies should be conducted to explore the influence of COVID-19 on people’s sorting behavior in the future.

## Figures and Tables

**Table 1 ijerph-19-06528-t001:** Publicity and education questions.

Independent Variable	Questions
Publicity and education [37]	Q5: My community has launched a publicity campaign for household waste classification.
	Q6: The household waste classification campaign can guide me in the correct classification of household waste.
	Q7: Regular publicity of household waste classification promotes my correct classification of household waste.
	Q8: A variety of waste classification publicity activities to promote my correct classification of household waste.

**Table 2 ijerph-19-06528-t002:** Classification criteria questions.

Independent Variable	Questions
Classification criteria	Q9: I think the current household waste classification standard is reasonable.
	Q10: I think the current household waste classification standard is simple and easy to understand.
	Q11: I think the unification of the household waste classification standard is helpful for daily classification.

**Table 3 ijerph-19-06528-t003:** Reward and punishment questions.

Independent Variable	Questions
Reward and punishment measures [38,39]	Q12: My community has reward and punishment measures for household waste classification as required.
	Q13: If there are incentives for household waste classification, I will be willing to classify.
	Q14: I will be penalized if I do not conduct household waste classification, I will be willing to classify.
	Q15: If the household waste classification implements charge by volume, I will be willing to classify.

**Table 4 ijerph-19-06528-t004:** Classification and supporting facilities questions.

Independent Variable	Questions
Classification of supporting facilities [40]	Q16: There are household waste classification collection facilities in my community.
	Q17: There are eye-catching classification standard descriptions on the household waste classification facility in my community.
	Q18: The household waste classification collection facility in my community is convenient for me to dispose of household waste.
	Q19: Intelligent waste classification equipment can attract me to carry out household waste classification.

**Table 5 ijerph-19-06528-t005:** Recycling system questions.

Independent Variable	Questions
Recycling system	Q20: The cleaning staff in my community sorts and recycles the classified waste.
	Q21: Classified and transported waste in our community sanitation department.
	Q22: Sort and dispose of sorted waste in our community sanitation department.
	Q23: The recycling norms in our community will prompt me to sort waste.

**Table 6 ijerph-19-06528-t006:** Subjective norm questions.

Independent Variable	Questions
Subjective norms [41]	Q24: My family supports the sorting of household waste.
	Q25: My friends all think I should sort household waste.
	Q26: Sorting household waste by others in my community will motivate me to sort.
	Q27: I think I should be consistent with the people around me.

**Table 7 ijerph-19-06528-t007:** Environmental knowledge questions.

Independent Variable	Questions
Environmental knowledge [42]	Q28: I know the categories of various household wastes.
	Q29: I know what the recyclable waste includes.
	Q30: I know to separate organic perishable waste from other waste
	Q31: I know which classification waste bin should be put into after the household waste classification.

**Table 8 ijerph-19-06528-t008:** Environmental attitude questions.

Independent Variable	Questions
Environmental attitude [43,44]	Q32: I think household waste should be sorted.
	Q33: I think household waste classification is beneficial to resource recycling and energy saving.
	Q34: I think sorting household waste is a responsible behavior.
	Q35: I think household waste classification can reduce pollution and protect the environment.

**Table 9 ijerph-19-06528-t009:** Classification behavior questions.

Dependent Variable	Questions
Classification behavior [45]	Q36: I will separate waste cardboard.
	Q37: I will separate kitchen waste.
	Q38: I will separate waste batteries and electronic equipment into categories.
	Q39: I will separate waste plastics.
	Q40: I will separate medicine waste.
	Q41: I will separate the scrap metal.
	Q42: I will separate the waste glass products.

**Table 10 ijerph-19-06528-t010:** Basic sample statistics.

Demographic Variable		Number (Persons)	Percentage (%)	Cumulative Percentage (%)
Gender	Male	322	51.7	51.7
	Female	301	48.3	100.0
Age	Under 18	20	3.2	3.2
	18–35	178	28.6	31.8
	35–55	314	50.4	82.2
	Older than 55	111	17.8	100.0
Education level	Senior high school or below	79	12.7	12.7
	Junior college	174	27.9	40.6
	Bachelor’s	283	45.4	86.0
	Master’s and above	87	14.0	100.0
Occupation	Student	74	11.9	11.9
	Self-employed businesses	80	12.8	24.7
	Education workers	111	17.8	42.5
	Government staff	54	8.7	51.2
	Retired workers	39	6.3	57.5
	Company employees	249	40.0	97.5
	Other	16	2.5	100.0

**Table 11 ijerph-19-06528-t011:** Results of confidence analysis.

Variable		Number of Items	Cronbach’s α
Governmental factors	Publicity and education	4	0.939
	Sorting standards	3	0.811
	Reward and punishment measures	4	0.903
Social factors	Auxiliary facilities for sorting	4	0.957
	Recycling system	4	0.866
Resident factors	Subjective regulation	4	0.890
	Environmental knowledge	4	0.863
	Environmental attitude	4	0.925
Waste-sorting behavior		7	0.904

**Table 12 ijerph-19-06528-t012:** KMO and Bartlett’s test for the influencing factor questions [47].

Kaiser–Meyer–Olkin Measure of Sampling Adequacy		0.873
**Bartlett’s test of sphericity**	Approximate chi-squared	4932.868
	Df	149
	Sig.	0.000

**Table 13 ijerph-19-06528-t013:** Results of the test for differences in waste-sorting behavior between the male and female respondents.

	Gender	Number of Respondents	Mean	Standard Deviation	t	Significance (Two-Tailed)
Waste-sorting behavior	Male	322	3.437	0.920	−2.574	0.010
	Female	301	3.616	0.806		

**Table 14 ijerph-19-06528-t014:** Results of the test for differences in waste-sorting behavior between respondents of different ages.

Age	Number of Respondents	Mean	Standard Deviation	F	Significance
Under 18	20	3.5865	1.20440	1.521	0.208
18–35	178	3.4198	0.95018		
35–55	314	3.5891	0.77108		
Older than 55	111	3.4929	0.92677		
Total	623	3.5235	0.87055		

**Table 15 ijerph-19-06528-t015:** Results of the test for differences in waste-sorting behavior between respondents with different education levels.

Education Level	Number of Respondents	Mean	Standard Deviation	F	Significance
Senior high school or below	79	3.320	0.946	7.644	0.000
Junior college	174	3.425	0.888		
Bachelor’s	283	3.527	0.871		
Master’s and above	87	3.893	0.630		
Total	623	3.523	0.871		

**Table 16 ijerph-19-06528-t016:** Results of the test for differences in waste-sorting behavior among respondents of different occupations.

Occupation	Number of Respondents	Mean	Standard Deviation	F	Significance
Student	74	3.473	0.957	1.952	0.071
Self-employed businesses	80	3.648	0.818		
Education workers	111	3.376	0.857		
Government staff	54	3.603	0.859		
Retired workers	39	3.234	1.091		
Company employees	249	3.581	0.832		
Other	16	3.705	0.637		
Total	623	3.523	0.871		

**Table 17 ijerph-19-06528-t017:** Correlation analysis results.

Variable	Mean	St. dev.	1	2	3	4	5	6	7	8	9
Publicity and education	3.321	1.029	1								
Sorting standards	3.541	1.029	0.446 **	1							
Reward and punishment measures	3.384	1.003	0.457 **	0.516 **	1						
Auxiliary facilities for sorting	3.284	1.105	0.542 **	0.446 **	0.515 **	1					
Recycling system	3.575	0.951	0.461 **	0.566 **	0.540 **	0.393 **	1				
Subjective regulation	3.667	0.958	0.374 **	0.395 **	0.392 **	0.374 **	0.447 **	1			
Environmental knowledge	3.687	0.912	0.380 **	0.403 **	0.388 **	0.365 **	0.427 **	0.585 **	1		
Environmental attitude	3.775	1.095	0.367 **	0.405 **	0.382 **	0.404 **	0.348 **	0.493 **	0.504 **	1	
Sorting behavior	3.524	0.871	0.522 **	0.548 **	0.562 **	0.508 **	0.525 **	0.515 **	0.509 **	0.477 **	1

** A 0.01 significance level (two-tailed), indicating a significant correlation.

**Table 18 ijerph-19-06528-t018:** Model summary.

Model	R	R^2^	Adjusted R^2^	Standard Error of Estimation
1	0.732	0.535	0.529	0.5974

Predicted variables: (constant), environmental attitude, recycling system, auxiliary facilities for sorting, environmental knowledge, publicity and education, sorting standards, subjective regulation, and reward and punishment measures.

**Table 19 ijerph-19-06528-t019:** ANOVA results.

Model		Sum of Squares	Degree of Freedom	Mean Square	F	Significance
1	Regression	252.261	8	31.533	88.356	0.000
	Residuals	219.125	614	0.357		
	Total	471.386	622			

Dependent variable: sorting behavior; predicted variables: (constant), environmental attitude, recycling system, auxiliary facilities for sorting, environmental knowledge, publicity and education, sorting standards, subjective regulation, and reward and punishment measures.

**Table 20 ijerph-19-06528-t020:** Table of regression coefficients.

Model		Non-Standard Coefficient		Standard Coefficient	t	Significance
		B	Standard Deviation	Beta		
1	(constant)	0.416	0.123		3.395	0.001
	Publicity and education	0.121	0.030	0.143	4.049	0.000
	Sorting standards	0.131	0.031	0.155	4.248	0.000
	Reward and punishment measures	0.157	0.032	0.181	4.916	0.000
	Auxiliary facilities for sorting	0.079	0.028	0.100	2.783	0.006
	Recycling system	0.082	0.034	0.089	2.395	0.017
	Subjective regulation	0.122	0.033	0.134	3.669	0.000
	Environmental knowledge	0.116	0.035	0.122	3.319	0.001
	Environmental attitude	0.074	0.028	0.093	2.695	0.007

## Data Availability

If readers want, they can E-mail the authors for the data.

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
