# Peer review of "Influencing Factors on the Household-Waste-Classification Behavior of Urban Residents: A Case Study in Shanghai"

_ijerph, 2022, doi:10.3390/ijerph19116528_

Round 1

Reviewer 1 Report

Dear authors!

 I would like to note a high level of statistical processing of the results. In my opinion, the work deserves high praise and should be published, but there are small comments that the authors could take into account.

  1. What type of municipal waste collection (In-house service delivery, Competitive tendering, «Side-by-side” collection) is characteristic of the study area?
  2. Conclusion and Discussion. Are there special additional training programs for children in schools and kindergartens in separate waste collection?

Are there any financial incentives for residents in the case of separate collection? (E.g. reduced rent).

In addition, I think it would not be out of place to mention several works related to the use of such a methodological tool as a questionnaire in municipal waste management, for example:

  • Kurbatova, A.; Abu-Qdais, H.A. Using Multi-Criteria Decision Analysis to Select Waste to Energy Technology for a Mega City: The Case of Moscow. Sustainability2020, 12, 9828. https://doi.org/10.3390/su12239828
  • Coelho, L.G.; Lange, L.C.; Coelho, H. Multi-criteria decision making to support waste management: A critical review of current practices and methods. Waste Manag. Res.2017, 35, 3–28.

Author Response

I would like to note a high level of statistical processing of the results. In my opinion, the work deserves high praise and should be published, but there are small comments that the authors could take into account.

Comment 1: What type of municipal waste collection (In-house service delivery, Competitive tendering, «Side-by-side” collection) is characteristic of the study area?

Response 1: Thank you for your suggestions. Side-by-side collection is characteristic of the study area.

Comment 2: Conclusion and Discussion. Are there special additional training programs for children in schools and kindergartens in separate waste collection?

Response 2: According to your suggestions, we add special training programs for children in schools and kindergartens in separate waste collection. Please see Line 519-524, Page 16.

Comment 3: Are there any financial incentives for residents in the case of separate collection? (E.g. reduced rent).

Response 3: According to your suggestions, we have modified this in the text. Please see Line 491-497, Page 16.

Comment 4: In addition, I think it would not be out of place to mention several works related to the use of such a methodological tool as a questionnaire in municipal waste management, for example:

Kurbatova, A.; Abu-Qdais, H.A. Using Multi-Criteria Decision Analysis to Select Waste to Energy Technology for a Mega City: The Case of Moscow. Sustainability2020, 12, 9828. https://doi.org/10.3390/su12239828

Coelho, L.G.; Lange, L.C.; Coelho, H. Multi-criteria decision making to support waste management: A critical review of current practices and methods. Waste Manag. Res.2017, 35, 3–28.

Response 4: According to your suggestions, we have cited the two papers in the text. Please see Line 192-193, Page 4.

Reviewer 2 Report

The article is interesting. However, there are some issues that need to be fixed before publication. Below you can find some constructive comments which should help to improve the work.

Title; Should “a” after colon be capitalized?

We invite the author(s) to refine the abstract, esp. when mentioning results (line 32-33; are the effects statistically significant?).

Line 49; What do you mean by “normal activities and environmental hygiene”?

Line 76; Ref. no. 5 – specify where this study was conducted.

Line 84; What would be “more abundant publicity forms”?

Line 159-160; The author(s) named three studies (ref. no. 24, 25, and 26), but then named only two countries’ residents “respectively”. Please, correct.

Line 161; Which study exactly??

Line 180-181; Please, check the first sentence of section 3. I reckon “investigate” should replace “investigated”, “on” and “in the literature” is excessive.

Line 190-191; Similar to the previous comment. The beginning of the sentence is unclear.

Very important – explain in detail and clarify the questionnaire (form) and data analysis. Give more information on development of the questionnaire (who created it, what literature exactly was followed as reference, etc.), how were the respondents selected (who it was sent to, were the respondents selected randomly), who conducted it (the research team, an agency, or someone else), etc.

Tables 10 and 11; Why “gender” and “environmental factors” are in bold?

Consider moving (some) tables to appendices.

In the conclusion you should bring back more authors and say how your study confirms/challenges the findings of other studies. You have only stated your results, but have not relate it to the literature.

What are some of the limitations of the study? It must be added to the conclusion and discussion section.

Author Response

The article is interesting. However, there are some issues that need to be fixed before publication. Below you can find some constructive comments which should help to improve the work.

Comment 1:Title; Should “a” after colon be capitalized?

Response 1: Thank you for your suggestions,we have modified this in the text. Please see Line 3, Page 1.

Comment 2: We invite the author(s) to refine the abstract, esp. when mentioning results (line 32-33; are the effects statistically significant?).

Response 2: Thank you for your suggestions. Data analysis revealed that the findings were statistically significant. Please see Line 21-31, Page 1.

Comment 3: Line 49; What do you mean by “normal activities and environmental hygiene”?

Response 3: Thank you for your suggestions,we have modified this in the text. Please see Line 51-52, Page 2.

Comment 4: Line 76; Ref. no. 5 – specify where this study was conducted.

Response 4: According to your suggestions. This study was conducted in Beijing, China. We have modified this in the text. Please see Line 86-87, Page 2.

Comment 5: Line 84; What would be “more abundant publicity forms”?

Response 5: Thank you for your suggestions. We modify it to “… more various forms of publicity” . Please see Line 95, Page 2.

Comment 6: Line 159-160; The author(s) named three studies (ref. no. 24, 25, and 26), but then named only two countries’ residents “respectively”. Please, correct.

Response 6: Thank you for your suggestions. The first are Mexican, the last two are Iranian, we have removed the respective.Please see Line 173, Page 4.

Comment 7: Line 161; Which study exactly??

Response 7: Thank you for your suggestions. The study refers to the above three studies.Please see Line 173, Page 4.

Comment 8: Line 180-181; Please, check the first sentence of section 3. I reckon “investigate” should replace “investigated”, “on” and “in the literature” is excessive.

Response 8: Thank you for your suggestions, we have modified in the text. Please see Line 195-210, Page 4-5.

Comment 9: Line 190-191; Similar to the previous comment. The beginning of the sentence is unclear.

Response 9:Thank you for your suggestions, we have modified this in the text. Please see Line 216-233, Page 5.

Comment 10: Very important – explain in detail and clarify the questionnaire (form) and data analysis. Give more information on development of the questionnaire (who created it, what literature exactly was followed as reference, etc.), how were the respondents selected (who it was sent to, were the respondents selected randomly), who conducted it (the research team, an agency, or someone else), etc.

Response 10: According to your suggestions. We have added information about questionnaires and data analysis. The concise questionnaire formulation process is clarified below in this section. All questions are originated from previous literatures and being modified furtherly to fit into this research context. Totally 700 questionnaires were distributed through online and offline channels in the urban area of Shanghai city, which ended up with 517 valid electronic and 106 valid paper questionnaires. Online questionnaires were distributed through the www.wjx.cn, the most popular online questionnaire design, distribution and collection website which can reach the most representative respondents of this study. Offline paper questionnaires were mainly distributed to the old who have difficulties in accessing the internet. Several neighborhoods which were clustered with the old people in Shanghai urban districts were selected. To avoid sample bias, the number of questionnaires that were distributed to different districts was various according to the population size. The research team which was comprised of four people in university’s research lab conducted this survey. 2 month and a week were used for questionnaire pre-test, modification and official distribution. Please see Line 195-209, Page 4-5.

Comment 11: Tables 10 and 11; Why “gender” and “environmental factors” are in bold?

Response 11: Thank you for your suggestions. This is a mistake in our work. We have modified this in the text. Please see Line 305-331, Page 9-10.

Comment 12: Consider moving (some) tables to appendices.

Response 12: Thank you for your suggestions. In order to better reflect the completeness of the article, we decided not to put the table in the appendix.

Comment 13: In the conclusion you should bring back more authors and say how your study confirms/challenges the findings of other studies. You have only stated your results, but have not relate it to the literature.

Response 13: Thank you for your suggestions. We've added links to relevant literature. Please see Line 479-497, Page 16.

Comment 14: What are some of the limitations of the study? It must be added to the conclusion and discussion section.

Response 14: According to your suggestions. We've added content about article limitations. Please see Line 525-529, Page 16-17.

Reviewer 3 Report

The paper analyzed the factors influencing the household waste classification behavior of residents in Shanghai. Using the primary data from 623 respondents, the study applied statistical and econometric analyses to test the hypotheses. Results provide recommendations for further development of domestic waste classification in Shanghai.

In general, the paper is informative and interesting. However, there are still some flaws to be addressed to further improve the paper such as the originality, novelty of the findings, choice of terms, econometric modelling, and formatting. Specifically,

  1. Abstract, L20-24: What research methods are used in the study?
  2. Introduction: Discuss what has been done and what is still missing. Then, introduce what is the academic contribution of the paper.
  3. Method: The details are excellent. Except for the regression part. From Econometrics point of view, I'd rather want to read the regression model/equation as a basis for the analysis and hypothesis testing than reading in long texts.
  4. Discussion: This should be improved. What is the novelty of the findings relative to current literature? 
  5. Conclusion: This is too dragging to read. Divide the topics into paragraphs.
  6. Ethical Statement, Informed Consent. These should be added. Read the Author Guidelines for studies involving human subjects. 
  7. Minor Comments: 
    1. "Propaganda" has a negative connotation. "Publicity" may be used or "Information Dissemination"
    2. Tables should be improved. Horizontal lines should be added in some parts (Tables 10 & 11). 
    3. Avoid cutting words in the tables.

Author Response

The paper analyzed the factors influencing the household waste classification behavior of residents in Shanghai. Using the primary data from 623 respondents, the study applied statistical and econometric analyses to test the hypotheses. Results provide recommendations for further development of domestic waste classification in Shanghai.

In general, the paper is informative and interesting. However, there are still some flaws to be addressed to further improve the paper such as the originality, novelty of the findings, choice of terms, econometric modelling, and formatting. Specifically,

Comment 1: Abstract, L20-24: What research methods are used in the study?

Response 1: Thank you for your suggestions. This article uses literature analysis, questionnaire survey and data analysis. Please see Line 21-24, Page 1.

Comment 2: Introduction: Discuss what has been done and what is still missing. Then, introduce what is the academic contribution of the paper.

Response 2: Thank you for your suggestions, we have added the academic contribution in the text. Please see Line 66-73, Page 2.

Comment 3: Method: The details are excellent. Except for the regression part. From Econometrics point of view, I'd rather want to read the regression model/equation as a basis for the analysis and hypothesis testing than reading in long texts.

Response 3: According to your suggestions. We inserted the formula in the regression section. The multivariate regression equation can be expressed as:

Y is the dependent variable; X is the independent variables;  is the intercept and … are the estimated coefficients;  is the random error. Please see Line 399-403, Page 13.

Comment 4: Discussion: This should be improved. What is the novelty of the findings relative to current literature?

Response 4: Thank you for your suggestions, we have modified this in the text. We added innovative content to the article. Please see Line 519-524, Page 16.

Comment 5: Conclusion: This is too dragging to read. Divide the topics into paragraphs.

Response 5: Thank you for your suggestions. We have segmented our conclusions. Please see Line 474-526, Page 16.

Comment 6: Ethical Statement, Informed Consent. These should be added. Read the Author Guidelines for studies involving human subjects.

Response 6: Thank you for your suggestions . This study does not involve Ethical content. 

Comment 7: Minor Comments:

  1. ”Propaganda” has a negative connotation. “Publicity” may be used or “Information Dissemination”
  2. Tables should be improved. Horizontal lines should be added in some parts (Tables 10 & 11).
  3. Avoid cutting words in the tables.

Response 7: a. Thank you for your suggestions. We have Already edited “Propaganda” to “publicity”. Please see Line 88-93, Page 2.

  1. Thank you for your suggestions, we have modified this in the text.Please see Line 305-330. Page 9-10.
  2. Thank you for your suggestions, we have modified this in the text.

Round 2

Reviewer 3 Report

The authors carefully addressed all reviewer's comments and suggestions and made significant changes to improve the manuscript.

I am looking forward to read the published version of the manuscript.